# Generation and Characterization of HDV-Specific Antisera with Respect to Their Application as Specific and Sensitive Research and Diagnostic Tools

**DOI:** 10.3390/v17091220

**Published:** 2025-09-07

**Authors:** Keerthihan Thiyagarajah, Sascha Hein, Jan Raupach, Nirmal Adeel, Johannes Miller, Maximilian Knapp, Christoph Welsch, Mirco Glitscher, Esra Görgülü, Philipp Stoffers, Pia Lembeck, Jonel Trebicka, Sandra Ciesek, Kai-Henrik Peiffer, Eberhard Hildt

**Affiliations:** 1Research Group, Paul-Ehrlich-Institute, 63225 Langen, Germany; keerthihan.thiyagarajah@pei.de (K.T.);; 2Medical Clinic 1, University Hospital, Goethe University Frankfurt, 60329 Frankfurt am Main, Germany; 3Division Allergologie—Zentrale Methodenentwicklung, Paul-Ehrlich-Institute, 63225 Langen, Germany; 4Infectious Diseases Unit, I. Department of Medicine, University Medical Center, Hamburg-Eppendorf, 20038 Hamburg, Germany; 5Department of Hepatology and Gastroenterology, Campus Virchow-Klinikum (CVK) and Campus Charité Mitte (CCM), Charité—Universitätsmedizin Berlin, 13435 Berlin, Germany; 6Hasso-Plattner-Institute Digital Health Cluster, University Potsdam, 14471 Potsdam, Germany; 7Division Gastroenterology, University Hospital Münster, 48147 Münster, Germany; pia.lembeck@pei.de (P.L.); kai-henrik@ukmuenster.de (K.-H.P.); 8Institute of Medical Virology, University Hospital Frankfurt, 60329 Frankfurt am Main, Germany

**Keywords:** hepatitis D virus, B-cell epitope, polyclonal antibodies, quantitative ELISA

## Abstract

The hepatitis D virus (HDV) is a small, defective RNA virus that induces the most severe form of viral hepatitis. Despite its severity, HDV infections are under-diagnosed due to non-standardized and costly diagnostic screening methods. However, limited research has been conducted on characterizing HDV-specific antibodies as alternative tools for diagnosis. Thus, we generated HDV-specific, polyclonal antibodies by immunizing rabbits with the HDV protein, small hepatitis delta antigen (SHDAg), in its oligomeric or denatured form. We identified SHDAg-specific linear epitopes by peptide array analysis and compared them to epitopes identified in HDV-infected patients. Using in silico structural analysis, we show that certain highly immunogenic domains in SHDAg, such as the coiled-coil domain, are masked in the oligomeric conformation of the protein; others, such as the second arginine-rich motif, are exposed. The nuclear localization signal is presumably exposed only by specific interaction of oligomeric HDAg with the HDV-RNA genome. Through surface plasmon resonance analysis, we identified two polyclonal antibodies derived from rabbit antisera with affinities in the lower nanomolar range. These antibodies were used to establish an ELISA that can quantitatively detect HDV virions in vitro and upon further optimization could be used as a promising alternative diagnostic screening method.

## 1. Introduction

The hepatitis D virus (HDV) is a human pathogenic RNA virus inducing severe viral hepatitis and an increased risk for the development of hepatocellular carcinoma (HCC) [1,2]. The single-stranded RNA genome of HDV encodes only for one protein, termed hepatitis delta antigen (HDAg), which in turn is expressed as two isoforms, a small (S) and a large (L) HDAg variant [3]. Both isoforms have distinct functional domains, including a coiled-coil domain (CCD) mediating oligomerization of HDAg [4], several RNA binding sequences mediating HDV genome binding, including an N-terminal RNA binding domain (RBD) [5] and two arginine-rich motifs (ARM) located in the central region of the protein [6], and a nuclear localization signal (NLS) for nuclear import [7]. The C-terminally elongated LHDAg variant has an additional nuclear export signal for shuttling between the nucleus and cytoplasm [8] and a farnesylation site, which is necessary for virus morphogenesis [9]. HDV is a defective satellite virus of hepatitis B virus (HBV). It naturally only occurs in coinfection with HBV, exploiting the surface proteins of HBV for envelopment and subsequent viral release [10,11].

According to two different meta-analyses, the global prevalence of HDV ranges between 0.16% and 0.8% of the general population [12,13]. This high discrepancy is due to the fact that no routine screening is implemented in many areas with high HBV prevalence, leading to under-diagnosis of HDV infections [14]. HDV diagnosis is predominantly performed by detection of HDV-RNA using reverse transcription polymerase chain reaction (RT-qPCR). However, this method is challenging due to the high sequence variability and complex secondary structure of the HDV-RNA genome and the dearth of standardization between diagnostic labs and, moreover, costly since it needs elaborate equipment. Therefore, RT-qPCR-based diagnostic screening methods are often times not easily applicable in resource-poor areas. As a result, HDV-infected patients are frequently not identified properly and not treated accordingly [15,16]. Novel cost-efficient and easily accessible approaches are urgently needed for a more extensive and broader HDV diagnosis. In this regard, we aimed to establish a diagnostic method based on the cost-efficient and easily accessible ELISA method using highly sensitive HDAg-specific antibodies, enabling the detection of HDAg in HDV particles as a surrogate marker for an active HDV infection. However, at present the structure of HDAg is not fully characterized. Likewise, limited research is conducted on anti-HDAg antibodies as diagnostic and therapeutic tools, presumably due to their lack of neutralizing effects in chronically HDV-infected patients [17].

Previous experiments on truncated HDAg variants indicated dimerization and subsequent multimerization of HDAg via its N-terminal oligomerization domain. Moreover, a recent study from 2025 revealed that the C-terminal region of HDAg consists of two disordered domains separated by a structured Helix-loop-helix motif. The Helix-loop-helix motif, including both ARM regions, mediates interaction with the HDV-RNA genome, leading to the formation of the ribonucleoprotein complex (RNP), which resembles nucleosome-like structures [18,19,20]. The intrinsically disordered domains, on the other hand, presumably enable HDAg to interact with a vast amount of host proteins by adapting its conformation. Through these interactions, the viral HDAg protein potently dysregulates cellular processes, such as cell signaling cascades, leading to disease progression [21,22].

In this light, highly sensitive HDAg-specific antibodies could not only be used as diagnostic tools but would thus also facilitate a more detailed analysis of the HDV replication cycle on a cellular level. Hence, we developed such antibodies by immunization of rabbits with native and denatured HDAg. The antibodies were characterized regarding their epitope recognition and kinetics. Potential target epitopes for the generation of therapeutic antibodies were identified. Moreover, this knowledge was implemented in the establishment of an HDV HDAg-specific ELISA, which enabled the quantitative detection of HDAg and ultimately viral titers in HDV virion samples produced in vitro and, to a certain extent, in HDV-infected patient sera. Our study provides the basis and points out factors that need to be considered for developing therapeutic antibodies to neutralize HDV infections and, moreover, points out factors that need to be optimized for developing a cost-efficient, ELISA-based screening method, which does not rely on complex and expensive equipment and thus can be implemented even in resource-poor areas.

## 2. Materials and Methods

### 2.1. Plasmid Construction

To obtain the expression vector pET21a(+)-SHDAg-His_6_, the SHDAg sequence was amplified using the plasmid pSVLD3 encoding a cDNA clone of HDV genotype 1 genome as a template [23] (accession number: AJ000558.1) and using the Q5 DNA polymerase (NEB). NdeI- and NotI-specific restriction sites were introduced into the flanking regions of SHDAg during amplification using primers 1 and 2 (Table 1). The amplified SHDAg sequence was then cloned into the bacterial expression vector pET21a(+) using the abovementioned restriction enzymes (NEB) and T4 DNA ligase (Thermo Scientific, Schwerte, Germany). By excluding the stop codon during amplification, the His_6_-tag encoded on the pET21a(+) vector was arranged at the C-terminus of the SHDAg open reading frame. For construction of pET21a(+)-SHDAg-Strep-tag II, the SHDAg-StII insert was amplified using primers 1 and 3 (Table 1)and the previously described construct pCDNA 3.1(-)-SHDAg-StII [22] as a template.

### 2.2. Protein Production and Purification

Recombinant SHDAg-His_6_ as well as SHDAg-StII were expressed in *E. coli* BL21(DE3) under the control of a lac operon by heat-shock transformation of the bacterial expression vectors pET21a(+)-SHDAg-His6 and pET21a(+)-SHDAg-StII. Upon induction with 1 mM IPTG, bacteria were grown at 37 °C for 4 h. Pelleted SHDAg-His_6_-expressing cells were resuspended in 30 mM Tris-HCl, 300 mM NaCl, pH 8.5, while pelleted SHDAg-StII-expressing cells were resuspended in PBS + 2 M NaCl, pH 7.1. Both buffers were supplemented with 25 µg/mL leupeptin, 1 mM PMSF, 20 µg/mL Pepstatin, 10 µg/mL Aprotinin, and 12.5 units/mL Benzonase, and lysed either by French press using 80–100 MPa or by sonification. Cell lysates were clarified by centrifugation (30 min at 4 °C and 48,298× *g*) and subsequent filtration using a 0.45 µm filter. Clarified SHDAg-His_6_ cell lysate was loaded on a pre-equilibrated 1 mL Histrap HP column (Cytvia). The column was washed with 10 CV of wash buffer (30 mM Tris-HCl, 300 mM NaCl, 60 mM imidazole, pH 8.5) and SHDAg-His_6_ was subsequently eluted with 10 CV elution buffer (30 mM Tris-HCl, 300 mM NaCl, 250 mM imidazole, pH 8.5). Clarified SHDAg-StII lysate was loaded on a pre-equilibrated 1 mL Streptactin HP column (Cytvia). SHDAg-StII was eluted using PBS + 2 M NaCl + 2.5 mM D-desthiobiotin, pH 7.1. Both antigen preparations were concentrated using an Amicon Ultra centrifugal filter with 10 kDa MWCO (Merck) and subsequently incubated with 5 mM MgCl_2_ and 80 U/mL benzonase for 30 min at room temperature. Then, the SHDAg-His_6_ preparation was dialyzed in 30 mM Tris-HCl, 300 mM NaCl, 5% glycerol, pH 8.5. Dialyzed SHDAg-His_6_ was centrifuged for 10 min at 10,000× *g* and the clarified supernatant was aspirated to obtain the native SHDAg-His_6_ protein variant. The precipitated fraction of purified SHDAg-His_6_ was further denatured with 6 M urea to obtain the denatured SHDAg-His_6_ form. The SHDAg-StII preparation, on the other hand, was dialyzed in PBS + 0.5% Tween20 and directly used for subsequent surface plasmon resonance (SPR) analysis.

### 2.3. Analytical Size Exclusion Chromatography

For size exclusion chromatography (SEC) analysis, 500 µL of sample was loaded on a Superdex 200 increase 10/300 GL (Cytvia) equilibrated with dialysis buffer. Respective molecular weights of fractions within purified protein samples were determined based on a calibration curve using β-amylase (200 kDa), ovalbumin (43 kDa), cytochrome *c* (12.4 kDa), and aprotinin (6.5 kDa) as references. Proteins were detected at a wavelength of 280 nm.

### 2.4. Dynamic Light Scattering Analysis

Dynamic light scattering (DLS) experiments were carried out on a Zetasizer Nano-ZS device (Malvern, United Kingdom). Then, 100 µL of respective sample was measured in a 3 mm quartz cuvette. Absolute size and number distributions were averaged by three measurements. Each measurement was based on 10 scans.

### 2.5. Structural Prediction

All structural predictions were performed using Alphafold 3 [24]. For prediction of native, oligomeric SHDAg-His_6_, 8 copies of the SHDAg-His_6_ protein sequence were used as an input sequence. The 394 nt HDV-RNA snippet was derived from the HDV-RNA genome sequence (accession number: AJ000558.1), in particular the nucleotide sequence from 197 (5′) to 1481 (3′), as previously described by Yang et al. [20].

### 2.6. Immunization

Two rabbits were immunized and boosted with the two different SHDAg-His_6_ preparations, native and denatured, in 8-week intervals. In total, antisera of each rabbit was taken four times after initial and booster immunization. Maintenance, immunization, and harvest of SHDAg-His_6_-specific antisera was carried out by Seramun Diagnostica GmbH (Heidesee, Germany).

### 2.7. Cell Cultivation and Transfection

Huh7, a human hepatoma cell line initially described in [25], was cultivated in Dulbecco’s Modified Medium (DMEM) supplemented with 2 mM L-glutamine, 10% fetal calf serum (Bio & Sell, FBS.S0615, Feucht, Germany), 100 units/mL penicillin, and 100 µg/mL streptomycin. The cell line was maintained at 37 °C in a humidified atmosphere containing 5% CO_2_.

For all Western blot analyses, Huh7 cells were transfected with the HDV genome cDNA encoding plasmid pSVLD3 or the mock plasmid pUC18 and lysed at 7 days pt. For generation of HDV virion samples for the ELISA experiments, Huh7 cells were co-transfected with the pSVLD3 plasmid and the HBV genome encoding plasmids pUC18 HBV genotype A or D. Respective HDV virion-containing supernatants were harvested at day 3, 7, 10, 14, and 16 pt. Supernatant of HBV-mono-transfected cells served as a negative control. For all immunofluorescence analyses, cells were transfected with the previously described recombinant SHDAg-StII-encoding plasmid pCDNA3.1(-)-SHDAg-StII [22] and fixed 3 days pt.

### 2.8. SDS-PAGE and Western Blot

Transfected Huh7 cells were lysed using radioimmunoprecipitation assay (RIPA) lysis buffer (50 mM Tris, 150 mM NaCl, 0.1% (*w*/*v*) SDS, 0.5% (*w*/*v*) sodium deoxycholate, 1% (*v*/*v*) Triton X-100, pH 7.2). Cells were additionally sonicated for sufficient lysis. A Bradford assay (Thermo Scientific, Schwerte, Germany) of cell lysates and purified protein samples was performed according to the manufacturer’s instructions. Then, 45 µg of total cell lysate or 10 µg of purified protein sample were loaded per lane of a 12% SDS-polyacrylamide gel. The PageRuler Plus Prestained Protein Ladder (Thermo Scientific) was used as the molecular weight marker. SDS-PAGE-separated proteins were either stained with Coomassie brilliant blue or subsequently Western blotted. Blots were blocked with blocking buffer (10% (*w*/*v*) skim-milk powder (Carl Roth, Karlsruhe, Germany) dissolved in TBS + 0.05% Tween20 (TBS-T)). All SHDAg-His_6_-specific antisera were diluted 1:5000 in blocking buffer. The control antibody, monoclonal mouse anti-HDAg antibody (clone FD3A7, cell culture supernatant, Kerafast, Newark, NJ, USA), was diluted 1:3000. Blocked membranes were probed with respective antibody dilutions overnight at 4 °C. Signals were detected using donkey anti-rabbit HRP-linked (Sigma Aldrich, Göttingen, Germany) and sheep anti-mouse HRP-linked secondary antibodies (Cytvia, Freiburg, Germany), Immobilon Forte Western HRP-substrate (Merck Millipore), and an Amersham ImageQuant 800 (GE Healthcare, Freiburg Germany) system.

### 2.9. Immunofluorescence Microscopy

Transfected Huh7 cells were fixed by incubation with 4% paraformaldehyde (Sigma Aldrich), permeabilized with 0.5% (*v*/*v*) Triton-X-100 (Sigma Aldrich, Göttingen, Germany) in PBS, and blocked with blocking buffer (5% BSA (Carl Roth) in PBS-T (PBS supplemented with 0.05% Tween-20)). Subsequently, cells were probed with respective HDAg-specific antisera diluted 1:500 in blocking buffer or a 1:200 diluted monoclonal mouse anti-Strep-tag II antibody (IBA Lifesciences, Göttingen, Germany). Antisera probed samples were stained with a goat anti-rabbit IgG (H+L) cross-adsorbed secondary antibody conjugated with an AlexaFluor^®^ 488 dye and the mouse anti-Strep-tagII probed samples with a goat anti-mouse cross-adsorbed secondary antibody conjugated with an AlexaFluor^®^ 594 dye (Invitrogen, Heidelberg, Germany; 1:1000 dilution), together with 250 ng/mL 4′,6-diamidin-2-phenylindole (Carl Roth, Karlsruhe, Germany). Samples were subsequently mounted and imaged with the Leica SP8 confocal laser scanning microscope (CLSM; Leica) using a 63× magnification oil immersion objective (numerical aperture = 1.4). All images were recorded with an image size of 1024 × 1024 pixels, pinhole dimension of 1.0 airy units, and scan speed of 200.

### 2.10. Patient Sera and Ethics

The HDV-infected patient sera analyzed in this study were provided by the University Hospital in Frankfurt am Main, Germany, and by the University Medical Center Hamburg-Eppendorf in Hamburg, Germany. The study was approved by the respective local ethics committees (Medical Faculty of the Goethe University Frankfurt Beschluss No. E 117/13, Geschäfts No. 314/13; Ethikkommission der Ärztekammer Hamburg, Germany, project number PV4081). The study was performed in compliance with the provisions of the Declaration of Helsinki from the World Medical Association and good clinical practice (GCP) guidelines.

### 2.11. Peptide Microarray

Here, 15 amino acid long peptides covering the full length S or LHDAg sequence with 10 amino acids overlap were synthesized using a MultiPep2 (Intavis AG) according to the principle of Merrifield [26]. Reference sequences for HDV genotypes 1 (accession number: AJ000558) and 3 (accession number: AIR77028) were derived from HDVdb [27]. Synthesized peptides and DMSO controls were spotted onto adhesive foil (Lasertab LAT-34-747-1) with 15 mm × 15 mm dimensions as octuplicates using a slide spotting robot (Intavis AG, Tübingen, Germany). Resulting peptide arrays were blocked with blocking buffer (Tris-buffer + 0.05% Tween20 + 10% (*w*/*v*) skim-milk powder (Carl Roth, Karlsruhe, Germany)) and probed with HDAg-specific rabbit antisera overnight at 4 °C in a 1:1000 dilution in blocking buffer. Polyclonal rabbit-derived HBV-specific anti-HBcAg antibody (K46) served as a negative control, and commercially available monoclonal rabbit-derived HDV-specific anti-LHDAg antibody (HL1051) served as a positive control. Control antibodies were diluted in the same manner. Signals were detected using goat anti-rabbit IgG IRDye 680 RD secondary antibody (LI-COR). For epitope mapping of HDV-infected patient sera, slides were initially blocked with casein blocking buffer (10% (*v*/*v*) casein, 5% D-Saccharose in TBS-T buffer) and subsequently air-dried. Arrays were probed with respective patient sera overnight at 4 °C, prediluted 1:20 in casein blocking buffer. Signals were detected using goat anti-human IgG IRDye 680 RD secondary antibody (LI-COR, Lincoln, NE, USA), diluted 1:5000 in casein blocking buffer, and a LI-COR Odyssey CLx (LI-COR, Lincoln, United States) system. Fluorescence intensities (FI) were determined using the software Image Studio Lite (LI-COR, Lincoln, United States). Respective log_2_ fold changes (log_2_ (FC)) of rabbit-derived antisera were determined by calculating the binary logarithm of the average fluorescence intensities of octuplicates divided by the average intensity of the DMSO negative control. Z-scores of respective HDV-infected patient sera were determined by subtracting the absolute fluorescence intensities by the average fluorescence intensity of the entire peptide array and dividing the difference with the standard deviation of the fluorescence intensities of the entire peptide array for each patient serum.

### 2.12. Surface Plasmon Resonance

Binding kinetics of HDAg-specific antisera were determined by SPR analysis using a Biacore T200 system (Cytvia, Freiburg, Germany) and a single cycle approach with SHDAg-StII captured. Production and purification of SHDAg-StII is described in Section 2.2. The capture antibody, StrepMAB-Immo antibody (IBA Life sciences), was used to capture SHDAg-StII. The antibody was diluted in immobilization buffer (10 mM sodium acetate, pH 5.0) to a final concentration of 30 µg/mL and immobilized on a CM5 sensor chip (Cytvia) using an amine coupling kit (Cytvia, Freiburg, Germany). Purified SHDAg-StII (0.25 µM in PBS-T; contact time 30 s) was captured to the flow cells, resulting in a capture level of about 20 RU. A total of four antisera were tested by SPR analysis, namely, the antisera native 1, native 4, denat 1, and denat 4. Prior to SPR analysis, the HDAg-specific polyclonal antibodies within these four antisera were purified using a 1 mL protein G HP column (Cytvia, Freiburg, Germany), according to the manufacturer’s instructions. Upon binding of the antigen SHDAg-StII to the StrepMAB-Immo antibody, purified antibodies were applied to the flow cells with a serial dilution of 0 µM, 0.0512 µM, 0.128 µM, 0.32 µM, 0.8 µM, and 2 µM, a contact time of 200 s each, and a final dissociation time of 900 s. After each run, the CM5 chip was regenerated with regeneration buffer (50 mM NaOH, 1 M NaCl). Flow rate was constant for all experiments and settled to 30 µL/s. Kinetic parameters were determined using the Biacore T200 evaluation software and the classical 1:1 binding model.

### 2.13. Purification of Polyclonal HDAg-Specific Antibodies

Native 4 and denat 4, being the most affine antibodies according to SPR analysis, were subsequently used as capture and detection antibodies, respectively, in the HDV-specific ELISA. To this end, denat 4 and native 4 antiserum were purified by affinity chromatography using a SHDAg-II immobilized NHS-activated sepharose column (Cytvia, Freiburg, Germany). Antibodies were eluted with elution buffer (0.1 M glycine, 200 mM NaCl, pH 2.0) as 400 µL fractions into 125 µL neutralization buffer (2 M Tris-HCl, pH 8.0). Purified antibodies were dialyzed to PBS. Then, purified denat 4 antibodies were biotinylated using a 40-fold molar excess of EZ-Link^TM^-NHS-Biotin (Thermo Scientific, Schwerte, Germany) according to the manufacturer’s instructions.

### 2.14. RT-qPCR

HDV-RNA was extracted from HDV-infected patient sera using the High Pure viral nucleic acid kit (Roche Diagnostics) and RT-qPCR performed using the Lightcycler Multiplex RNA Virus master and the Lightmix Modular HDV (Roche Diagnostics, Mannheim, Germany) in a white lightcycler 480 Multiwell plate (Roche Diagnostics). HDV-RNA extraction and qPCR were performed according to manufacturer’s instructions. All samples were measured in duplicates. Final HDV copy numbers were interpolated from a standard curve.

### 2.15. ELISA Sample Preparation

HDV-infected patient sera, supernatant samples, as well as serial dilutions of protein standard were centrifuged. The aqueous phase was aspirated and mixed with NP40 to a final concentration of 1% (*v*/*v*). Samples were then incubated for 5 min at 55 °C, again centrifuged, and supernatant was transferred into a new reaction tube. HDV-infected patient samples were additionally mixed with 1% BSA (*w*/*v*) in TBS-T after final centrifugation.

### 2.16. ELISA Protocol

High-binding 96-well plates (Biozol, Hamburg, Germany) were coated with 0.5 µg/mL purified polyclonal native 4 antibody diluted in coating buffer (137.14 mM sodium bicarbonate, 62.86 mM sodium carbonate, pH 9.6) overnight at 4 °C. Coated plates were probed with 100 µL of respective pretreated samples in duplicates. For absolute quantification, a serial dilution of purified SHDAg-His_6_ was probed alongside the samples. The SHDAg-His_6_ standard was diluted in 5% BSA in TBS-T to the final concentrations of 8, 4, 2, 1, 0.5, 0.25, 0.125, and 0.0625 ng/mL. Purification of SHDAg-His_6_ is described in Section 2.1. As a negative control, all plates were probed with supernatant samples of HBV genotypes A and D mono-transfected cells.

All samples and standards were incubated for 1.5 h at RT and subsequently incubated at 4 °C overnight with constant agitation. Probed plates were incubated with 0.5 µg/mL biotinylated denat 4 antibody and subsequently with 1:500 diluted streptavidin-HRP antibody (Sigma-Aldrich, Grünwald, Germany), diluted 1:500 each for 1 h at RT and prediluted in 5% BSA in TBS-T. Between each step, plates were washed three times with 300 µL TBS-T. Finally, plates were incubated with TMB substrate for 30 min, followed by addition of 0.5 M H_2_SO_4_ and absorbance detection at 450 nm using a Tecan spark plate reader (Tecan, Crailsheim, Germany). A standard curve based on the purified SHDAg-His_6_ dilution series was generated by applying the 4PL sigmoidal fit. The standard curve ranging from 0.0625 to 8 ng/mL defined the quantifiable range. The HBV virion-containing negative control elicited absorbance values below 0.1 mAU. Thus, the threshold for positive signals was defined by a 450 nm absorbance value above 0.1 mAU. Accordingly, final HDAg concentrations of positive samples were interpolated based on the standard curve.

All relevant plasmid sequences, protein sequences, and RNA sequences used in this study and the primary data of the figures are available at the following link to Mendeley Data: https://data.mendeley.com/preview/h7s2kvrm66?a=abfaa40a-b826-446e-bc1d-a2c8fb988aff (accessed on 20 June 2025).

## 3. Results

### 3.1. Recombinant SHDAg Preserves Dynamic Multimeric Conformation Under Native Purification Conditions

To obtain HDV-specific antibodies, two rabbits were immunized with the recombinant SHDAg-His_6_ protein either in its native or denatured form. SHDAg-His_6_ was initially expressed in *E. coli* and purified under native conditions using Ni-NTA. The purified SHDAg-His_6_ was approximately 80% pure. As observed in the Western blot and Coomassie stain, minor contaminants at 15 kDa were present after purification. These contaminants were attributed to N-terminally degraded SHDAg-His_6_ since the respective protein fragment was reactive to the monoclonal His_6_ antibody (Figure 1A). We further analyzed the purified SHDAg-His_6_ using SEC and DLS. SEC analysis revealed two major peaks corresponding to apparent molecular weights of 100 and 200 kDa, corresponding to SHDAg tetramers or octamers, respectively. A small fraction of the native protein migrated in the void volume, indicating either higher-order oligomerization or partial aggregation of the purified protein. Additionally, two major peaks at 18 mL and 20 mL were observed, corresponding to 10 kDa and 5 kDa, respectively, reflecting the degradation products observed in the Western blot analysis. According to the SEC analysis, the purified sample contained only oligomeric or degraded SHDAg-His_6_, not monomeric or dimeric SHDAg-His_6_ (Figure 1B). Moreover, DLS analysis also confirmed the presence of oligomers in the purified samples. Two major populations with an average size ranging from 10 to 50 nm and from 100 to 400 nm were detected, while no population below 10 nm could be detected (Figure 1C). The number plot further revealed that the largest population of protein consisted of an average size ranging from 10 to 20 nm and that protein aggregates only accounted for a proportion of less than 0.1% within the purified sample (Figure 1D). To further investigate the conformation of the purified protein sample, a structural prediction of homo-octameric SHDAg-His_6_ was performed using Alphafold 3. The RBD as well as the CCD could be predicted with very high confidence, as reflected by a predicted local distance difference test (pLDDT) value above 90, while the first and second ARMs resulted in confident prediction (90 > pLDDT >70). However, the pLDDT for the structural prediction of the NLS and the C-terminus was very low.

In general, the prediction revealed a concentric alignment of the monomers with a 5 nm hole formed by the respective alpha-helically structured RBDs. The CCDs of adjacent SHDAg-His_6_ monomers interact with each other in an antiparallel coiled-coil arrangement. The prediction also revealed a helix-loop-helix motif within the first and second ARMs (Figure 1E). The distance between the most distant alpha helices that could be predicted with confidence was approximately 13 nm, which is in line with the approximate size of purified SHDAg-His_6_ according to the DLS analysis. Conclusively, purified SHDAg-His_6_ was indeed in its native oligomeric conformation. Treatment with urea resulted in disruption of the oligomer (data not shown). Thus, two different rabbits were solely immunized either with the native or denatured protein preparation to obtain antibodies that are only specific to either the native, oligomeric SHDAg-His_6_ or the denatured, monomeric SHDAg-His_6_ protein.

Additionally, to model the interaction between the multimeric SHDAg-His_6_ and the HDV-RNA genome, a 394 nt long HDV-RNA genome snippet, which was previously shown to be the minimal segment to form complexes with HDAg, was incorporated into the prediction. Interestingly, although the central hole formed by the RBD and CCD remained, the overall conformation downstream of the CCD changed upon inclusion of the RNA snippet. Specifically, both ARMs together with the NLS of adjacent SHDAg-His_6_ monomers formed holes, and the RNA snippet was threaded through these holes, being wrapped around the HDAg oligomer (Figure 1E,F). These predictions highlight the dynamic nature of SHDAg. We considered the implications of these structural predictions for the design of immunizations, as well as for all subsequent interpretations of antibody epitope mapping.

### 3.2. HDAg-Specific Antisera Show Major Differences in Their Binding Specificity

Antisera of the two immunized rabbits were collected at four different time intervals after booster immunizations, resulting in four distinct antisera specific to denatured SHDAg-His_6_, from now on referred to as “denat 1 to 4”, and four antisera specific to native SHDAg-His_6_, classified as “native 1 to 4”. Next, the reactivity of obtained antisera toward HDAg was analyzed. The binding specificity toward denatured HDAg was tested by Western blot analysis using denatured samples. A commercially available monoclonal antibody targeting both the S and LHDAg acted as a positive control. The Western blot analysis revealed two distinct signals at 25 kDa and 30 kDa, which were only specifically detected in the sampled derived from HDV genome cDNA transfected cells and not in the mock transfected control samples. Since these two signals were also detected by the commercially available antibody, the signals were designated as S (lower) and LHDAg (upper signal). Since S and LHDAg share the same 195 amino acids, antibodies targeting SHDAg should also bind to LHDAg. In line with that, our antisera also recognized both HDAg variants expressed in mammalian cells, like the control antibody. Interestingly, binding specificity of the different antisera varied. For instance, native 4 antisera showed the highest binding specificity with little to no unspecific binding, while denat 3 antiserum showed high unspecific binding (Figure 2A).

Next, the binding specificity toward the native HDAg protein was assessed. To this end, Huh7 cells expressing recombinant SHDAg with a C-terminal Twin-Strep-tag II (StII) were fixed with paraformaldehyde to preserve conformational epitopes and probed with respective antisera and a reference antibody. The reference antibody, specific to the StII, detected the recombinant protein mainly in the nucleus. Considering that native SHDAg is generally translocated into the nucleus via its NLS, this observation validated the structural integrity of the expressed recombinant protein. Moreover, the immunofluorescence analysis further revealed high overlap of the signals detected by the antisera with the signals of the reference antibody, indicating successful recognition of native SHDAg-StII by all antisera. However, similar to the Western blot analysis, the antisera showed different binding specificities. For instance, similar to the Western blot analysis, denat 3 showed high background signals when compared to the reference antibody (Figure 2B).

Taken together, these data suggest that the generated antisera are suitable for analyzing cellular systems that replicate HDV via Western blot and immunofluorescence microscopy with high specificity and sensitivity.

### 3.3. Distinct Immunogenic Epitopes Are Inaccessible During B-Cell Epitope Recognition in the Native Conformation of SHDAg

To further characterize the antisera with respect to the recognized linear B-cell epitopes, a peptide array consisting of 15 amino acid long peptides with an offset of 10 amino acids, covering the full-length LHDAg derived from HDV genotype 1, was designed. The peptide arrays were incubated with respective antisera, and recognized epitopes were detected with an IRDye 680RD conjugated anti-rabbit antibody. Strongly immunogenic linear epitopes were defined by a log_2_ fold change (FC) greater than 4. For validation, the peptide array was incubated with the polyclonal rabbit-derived antiserum K46, which targets the core protein of HBV. As expected, the negative control antibody did not recognize any epitopes, reflecting low unspecificity of the peptide array. As a positive control, peptide arrays were incubated with the commercially available rabbit-derived monoclonal antibody HL1051, which specifically targets the C-terminal LHDAg-specific extension. As expected, the HL1051 antibody specifically recognized the epitope “WDILFPADPPFSPQS” within the C-terminal LHDAg extension, confirming the functionality of the peptide array. Overall, several immunogenic epitopes within the main functional domains of HDAg were recognized by the antisera.

Interestingly, some epitopes were solely recognized by all denat but by none of the native antisera. In particular, the epitope “LKKIEDENPWLG” within the CCD, the epitope “KERQDHRRRKAL” within the first ARM, the epitope “GPPVGGVNPL” within the proline-glycine-rich sequence (PGRS), and the epitope “ESPFSRTGEGLDIRG” within the C-terminus of SHDAg were strongly recognized by all denat antisera, while the same epitopes were not recognized by any of the native antisera. Likewise, the epitope “VDSGPGKR”, located downstream of the NLS region, was recognized in two consecutive peptides only by all four denat antisera and none of the native antisera. Interestingly, the NLS region was recognized by all denat antisera as a single peptide bearing the sequence “EGAPPAKRARTDQME”. Only denat 3 antiserum also recognized the consecutive peptide. The epitope “GGREEILEQWAGRK” within the RBD was indeed recognized by the native antisera, however, to a lesser extent compared to the denat antisera. In contrast to that, the epitope “EDERRERRVA” within the second ARM was strongly recognized by the native antisera with an overall higher average log_2_(FC) compared to the denat antisera. Moreover, the epitope “LDIRGNQGFP” was recognized only by the native 1 antiserum, but not by any of the other native antisera. Further, the adjacent peptide sequence prior to this specific epitope was also recognized by denat antisera (Figure 3A,C).

To investigate the accessibility of the functional domains to B-cells, we superimposed the functional domains onto the Alphafold-3-based structural prediction of the octameric SHDAg-His_6_ protein. As can be seen in the prediction, the CCD, RBD, and first ARM are located in the center of the protein, near the central hole. In the oligomeric conformation of the protein, they are rather enclosed and likely less accessible from the outside. The second ARM, on the other hand, is located on the periphery of the protein and is, therefore, likely more accessible in the oligomeric state (Figure 3B). The NLS, PGRS, and C-terminus could not be predicted accurately, hence they are indicated as loops (Figure 1E and Figure 3B). Since these epitopes were not recognized by the native antisera, we hypothesize that they are masked in the oligomeric state of SHDAg.

### 3.4. HDAg-Specific Antisera Show Pan-Genotypic Binding Capacity

Next, we tested the pan-genotypic binding capability of the obtained antisera. Since HDV genotypes 1 and 3 have the least sequence similarity among the eight HDV genotypes, a peptide array based on the SHDAg sequence derived from HDV genotype 3 was generated. Interestingly, the HDV genotype-3-based peptide array recognized only the epitopes within the CCD and the C-terminus, as well as the epitope within the first ARM to some extent, while the epitopes within the RBD, NLS, second ARM, and PGRS were not recognized (Figure 4A). In accordance with the HDV genotype-1-based peptide array, only the denat antisera recognized these epitopes. Multiple sequence alignment of sequences representing all eight known HDV genotypes revealed that the epitopes within the CCD and the first ARM are highly conserved among all eight HDV genotypes, marking them as potential target epitopes for pan-genotypic antibodies. Conversely, the C-terminus is generally less conserved, though it shows high sequence similarity between genotypes 1 and 3.

### 3.5. The NLS of HDAg Is a Prominent Target B-Cell Epitope in Chronically HDV-Infected Patients

Previous experiments revealed the high immunogenicity of certain HDAg epitopes in immunized rabbits. However, B-cell epitope recognition in chronically HDV-infected patients can differ greatly from epitope recognition in rabbits with purified HDAg. Thus, the HDAg genotype 1 peptide arrays were probed with the sera of 18 chronically HDV-infected individuals with different clinical backgrounds in the following experiment. Since immune responses, and ultimately antibody titers, vary greatly depending on the patients’ clinical backgrounds, z-scores for the respective peptides were determined, and linear epitopes were defined as having a z-score greater than 1.5. In general, the patient sera recognized different epitopes than the rabbit antisera did. Unlike the denatured antisera of immunized rabbits, the highly immunogenic epitopes within the RBD and CCD were not recognized by any of the patient sera. Additionally, the epitope within the second ARM, which is specifically recognized by native rabbit antisera, was also not recognized by any of the patient sera. However, one patient serum recognized the epitope EEEEELRRLTEEEDERR, which is located upstream of the second ARM. Further, one patient recognized the epitope WDILFPADPPFSPQS within the LHDAg extension. Interestingly, nine out of eighteen tested patient sera reacted to the peptide “EGAPPAKRARTDQME” containing the NLS of HDAg, and four patients additionally recognized the consecutive peptide. All linear B-cell epitopes identified in immunized rabbit and HDV-infected patient sera are listed in Table 2. In general, the epitope within the NLS was identified as the most predominant and conserved epitope among all HDV-infected patient sera tested (Figure 5A). Structural comparison of the NLS region of oligomeric SHDAg-His_6_ either in interaction with a 394 nt HDV-RNA genome sequence, a random RNA sequence, or without any RNA interaction indicated major conformational changes of the NLS region, depending on the HDAg–RNA interaction. According to the Alphafold predictions, NLS regions of adjacent SHDAg monomers are attached to each other in the absence of RNA and in the presence of random RNA. The random RNA is threaded through the central hole of the HDAg oligomer, while the NLS regions are not involved in the protein–RNA interaction. In contrast to that, in the presence of the HDV-RNA genome snippet, the same adjacent NLS regions form a pit and the HDV-RNA genome snippet lays inside the pit, while the central hole remains unoccupied (Figure 5B–D).

### 3.6. HDAg-Specific Antisera Bind with High Affinity to the Target Antigen, Making Them Suitable for ELISA-Based HDV Detection

Epitope mapping of the HDAg-specific antisera revealed a broad epitope recognition of HDAg by the denat antisera and distinct recognition of the oligomeric HDAg by the native antisera. These properties are beneficial in immunoassays. However, high affinities of the antibody toward the target are also crucial. Thus, for analysis of the affinity of the antibodies, we determined the kinetic parameters of the antigen–antibody interaction. To this end, antibodies within the antisera were enriched by Protein G affinity chromatography and binding to purified SHDAg-StII was analyzed using SPR analysis. For comparison, antibodies obtained after initial immunization were compared to respective antibodies after the fourth booster immunization. In general, all enriched antibodies showed low association rates but also very low dissociation rates (Figure 6A). In particular, enriched denat 4 antibody had a lower dissociation rate than enriched denat 1 antibody, although the association rates were similar. Enriched native 1 antibody showed the fastest association rate with 1540 ± 248 1/Ms, while enriched native 4 antibody showed the lowest dissociation rate with 6.47 × 10^−5^ ± 3.00 × 10^−7^ 1/s. Accordingly, native 4 antibody had the lowest dissociation constant with 26 ± 3 nM, 6-fold lower than that of native 1 antibody. Denat 4 antibody had the second lowest dissociation constant with 51 ± 20 nM, 2-fold higher than that of denat 1 antibody (Table 3). Conclusively, the affinity of the antibodies majorly increased from the initial immunization to the fourth booster immunization.

Since affinities in the lower nanomolar range are suitable for immunoassays, next, a sandwich ELISA using the two antibodies with the highest affinity toward the HDAg protein, namely, native 4 and denat 4 antibodies, was established with the purpose of HDV virion detection in vitro and in vivo.

Due to the specificity toward oligomeric HDAg, which is the most likely conformation in infectious HDV virions, antigen-affinity purified native 4 antibody was used as the capture antibody. Antigen-affinity purified denat 4 antibody was used as the detection antibody due to its broad epitope recognition and to avoid displacement effects with the capture antibody. To exclude cross-reactivity with unspecific antibodies and other serum components, both the capture and detection antibodies were previously affinity purified using a SHDAg-StII cross-linked resin. The denat 4 antibodies were additionally biotinylated so that signals could be detected using horseradish peroxidase conjugated streptavidin and the colorimetric substrate TMB.

Initially, various blocking agents, including BSA, dry milk, and FCS, were tested as diluents. In this regard, 5% BSA showed overall the best signal-to-noise ratio at various capture antibody concentrations and was, therefore, chosen as the ELISA diluent for antibody dilution (Figure 6B). Next, the optimal antibody concentration was determined by performing a serial dilution ranging from 0.125 µg/mL to 1 µg/mL of the capture and biotinylated detection antibody. Overall, the S/N increased with the increasing capture and detection antibody concentration. At a 1 µg/µL capture antibody and detection antibody concentration, however, the S/N declined. Thus, a 0.5 µg/mL capture and detection antibody concentration was chosen as the appropriate antibody concentration (Figure 6C). In the next step, the capability of the ELISA to detect HDV virions was assessed. Therefore, supernatant samples of HDV and HBV genotypes A and D genome cDNA co-transfected cells, harvested at different time points, were probed. Supernatants of respective HBV mono-transfected cells served as a negative control. Both HDV-HBV co-transfected samples showed an increase in signal intensity over time, with a maximum signal intensity at day 14 pt and a slight decline at day 16 pt. In contrast, both HBV mono-transfected control samples showed a steady signal intensity over time. Overall, the signal intensity of these samples was below that of the HDV-HBV co-transfected samples, indicating specific detection of HDV virions in the supernatant with little to no cross-reactivity with HBV virions (Figure 6D). Finally, the capability of the ELISA to detect HDV virions quantitatively in HDV-infected patient samples was assessed, for further details see Appendix A. To validate the ELISA results, viral loads of respective samples were determined by RT-qPCR. The RT-qPCR analysis of the HDV-infected samples revealed different viral loads amongst all tested patients, ranging from as low as 1000 copies/mL up to almost 10^7^ copies/mL, confirming the presence of HDV virions in all 18 tested samples (Figure 6E). Next, the previously established ELISA was probed with all 18 HDV-positive samples. To quantify the obtained ELISA results, a serial dilution of purified SHDAg-His_6_ was probed alongside the HDV-infected patient samples and a standard curve was generated (Figure 6F). Two out of eighteen tested patient sera exhibited quantifiable absorbance values above the background signal. Accordingly, in patient 304, 0.4 ng/mL HDAg was detected, while in patient 7, 0.9 ng/mL HDAg was detected (Figure 6G). Respective viral loads determined by RT-qPCR were 186,779 copies/mL for patient 7 and 350,561 copies/mL for patient 304. Other highly HDV-positive samples with titers above 1 million copies/mL did not exhibit quantifiable HDAg amounts in the ELISA (Figure 6E). In conclusion, the ELISA is indeed capable of quantitative HDV virion detection in in vitro samples but needs further optimization for in vivo samples.

## 4. Discussion

Antiviral antibodies have proven to be efficient tools in viral research, diagnosis, and treatment. However, compared to other viruses, there is only limited knowledge about HDV-specific antibodies. Considering the high pathogenicity of HDV, limited treatment options, and the largely elusive replication cycle, thoroughly characterized HDV-specific antibodies are needed to overcome these issues. Moreover, regions in which HDV infections are highly endemic have a rudimentary medical infrastructure, and diagnostic methods, such as RT-qPCR, are not easily accessible. Further, HDV genome RNA levels do not necessarily correlate to the amount of infectious viral particles, since naked HDV-RNA can presumably also be released by disrupted infected cells. Thus, an alternative cost-efficient method to detect a different viral component is necessary to improve diagnosis and monitoring of HDV infection. To address these issues, we generated highly sensitive HDV HDAg-specific polyclonal antibodies and characterized said antibodies regarding binding kinetics and epitope recognition based on in silico structural analysis. We then correlated the B-cell epitope recognition of the antisera to that of HDV-infected patient sera and identified epitope sequences that are predominantly recognized by the human B-cell response. Finally, we established an ELISA that enables the quantitative detection of HDV virions produced in vitro and to a certain extent in HDV-infected patient sera. Upon further optimization, the ELISA has the potential to be implemented in clinics as an alternative diagnostic method.

Overall, more than 40% of the SHDAg sequence was recognized by the antisera derived from rabbits immunized with denatured HDAg. However, it has to be considered that the small fraction of degradation products observed in our SHDAg-His_6_ preparation could have affected epitope recognition and antibody specificity. Since HDAg degradation is also observed in the liver tissue of HDV-infected humans and woodchucks, protein degradation is most likely an intrinsic property of HDAg, presumably catalyzed by an autoproteolytic mechanism [28,29]. Therefore, the inevitable HDAg degradation in our antigen preparation likely resembles the natural infection process. Interestingly, most of the recognized linear epitopes, such as the ones in the CCD, the NLS, or both ARMs, were also described as reactive CD4- and CD8-positive T-cell epitopes in HDV-infected patients, emphasizing the high immunogenicity of these domains [30]. However, only the epitope within the second ARM was immunogenic in the rabbit immunized with the native SHDAg protein, likely due to steric inaccessibility of the other domains, such as the CCD or RBD in the oligomeric state of the protein, as reflected by our structural predictions. In line with that, a study by Bichko et al. investigating the epitopes exposed in the viral RNPs derived from HDV virions revealed that the CCD domain was not exposed in the viral RNPs even upon release of the HDV-RNA using vanadyl nucleoside complexes, indicating the persistent multimerization of the HDAg protein via the CCD even after depletion of the presumably stabilizing HDV-RNA. Our study further corroborated the previous observation of Bichko et al., that the CCD is indeed not exposed in the oligomeric conformation of HDAg despite being highly immunogenic and, moreover, that oligomerization of HDAg via the CCD is independent of the interaction of HDAg with HDV-RNA [31]. A very recent structural analysis by Yang et al. using NMR spectroscopy revealed that both ARMs indeed form a conserved helix-loop-helix motif, in line with our structural prediction. According to their study, the first ARM is fully integrated into the helix-loop-helix motif, while the second ARM extends beyond the structured motif, with amino acids R142–Gly146 being unstructured [20]. Therefore, it is conceivable that the second ARM, due to its unstructured C-terminus, is more flexible and thus also sterically more accessible in the oligomeric conformation of HDAg.

Furthermore, the comparison of the epitope mapping of immunized rabbit-derived antisera with that of HDV-infected patient-derived sera revealed only one overlapping epitope, while nine out of eighteen tested patient sera reacted to the peptide “EGAPPAKRARTDQME”. Similar to the rabbit-derived antisera, the consecutive peptide was only recognized in some cases, indicating that this specific peptide likely encompasses the entire linear epitope necessary for antibody binding and splitting of this peptide will affect epitope recognition. In this regard, one limitation of this study is that the peptides in the peptide array are synthesized with shifts of five amino acids between consecutive peptides. Smaller shifts would, therefore, be more insightful in this particular sequence region. Nevertheless, two previous studies also observed the immunodominance of this particular sequence within the NLS region in the B-cell response of HDV-infected individuals. In previous SPR imaging studies, 9 out of 17 HDV-infected patient sera showed the strongest reactivity toward this particular peptide sequence [32]. Likewise, in a study conducted by Wang et al., sera of five patients with high anti-HDV titers specifically reacted to this particular sequence within the NLS region. Interestingly, in the same study the patient sera also reacted to the LHDAg extension, similar to one of our patients’ serum [33]. Our structural prediction of oligomeric SHDAg-His_6_ with different RNAs indicated structural changes in the NLS region due to the protein–RNA interaction. Interestingly, prediction of SHDAg-His_6_ conformation with a 394 nt HDV-RNA genome segment, the minimal genome segment that is sufficient to form complexes with HDAg, revealed a distinct structural change in the NLS region, which presumably is unique to interaction with the rod-like shaped HDV-RNA sequence, since the same conformational change was not observed with random RNA. In this regard, a recent NMR spectroscopy analysis of truncated SHDAg with the same 394 nt HDV-RNA snippet revealed that the N-terminal region of the NLS region (Lys72–Arg75) is indeed involved in the interaction of HDAg with HDV-RNA [20]. This is in line with our peptide array analysis, in which the peptide encompassing the NLS region was not recognized in native antisera obtained by immunization with oligomeric HDAg expressed and purified in the absence of the HDV-RNA. However, the same region was predominantly recognized by sera of HDV-infected patients. Therefore, we hypothesize that the conformation of the NLS region is changed by specific interaction with the HDV-RNA genome, and that this HDV–RNA interaction-mediated conformational change makes the NLS region accessible to the humoral response in HDV-infected patients. This hypothesis needs further structural conformation but could reveal a novel mechanism of how shuttling of the HDAg protein between the nucleus and cytoplasm is regulated by either masking or exposing the NLS region through interaction of the protein with the unique RNA genome. In light of that, usage of HDAg-specific antibodies for either diagnosis or therapy is limited by the dynamic conformation of HDAg, which itself likely depends on the interaction with the HDV-RNA. Nevertheless, our HDV-specific ELISA can indeed detect HDV virions, as reflected by the viral release kinetics of in vitro produced HDV virions, which peaked at day 14 post-transfection. This is in line with a previous study in which a steady increase of HDV secretion by HBV HDV cDNA co-transfected cells, with a maximum secretion between days 10 and 13 and a slight decline after day 13 pt, was observed [34]. However, in HDV-infected patient sera, HDV was detected only in 2 out of 18 tested samples (see Table A1 in the Appendix A for demographic and clinical characteristics of all tested patients). Since these two particular patient samples only contain intermediate viral titers and samples with even higher samples were not detected, insufficient sensitivity of our ELISA can be ruled out. Several reasons could have affected HDV capture and detection in the ELISA, such as different HDAg conformations in the patient sera depending on the protein–RNA interaction, which might have led to masking of epitopes, as also observed in our epitope mapping. Other than that, although our polyclonal antibodies show overall high affinities, certain endogenous antibodies within the infected patient sera, such as the NLS region-specific antibodies, could have even higher affinities and, therefore, compete with the capture antibodies, which predominantly recognize the 2nd ARM and hence prevent antigen capture in the ELISA. Moreover, HDV infection is characterized by severe liver damage, presumably leading to release of non-infectious HDV-RNA or HDAg protein from disrupted cells. Thus, detection of serum HDV-RNA levels only does not necessarily reflect the actual infectious viral titer. Additionally, different treatment conditions could affect intrahepatic HDV-RNA levels and HDAg protein levels differently. Considering the limited availability and lack of standardization of HDV-RNA quantification assays between diagnostic centers, as well as only modest reproducibility of the HDV-RNA quantification assays, as shown by the external quality control studies [35], it is even more clear that serum HDAg levels need to be quantified alongside HDV-RNA levels as a complementary biomarker to precisely monitor HDV infectivity, treatment efficiency, and viral clearance in HDV-infected patients. We believe that this study provides a promising basis for developing a sensitive ELISA-based assay to quantify serum HDAg levels as a complementary marker alongside HDV-RNA levels in clinical samples.

## 5. Conclusions

In conclusion: our thorough analysis of HDV-specific B-cell epitopes confirmed findings of previous studies that many immunogenic domains of HDAg are masked in its oligomeric conformation, which could affect viral clearance by the humoral immune response in chronically infected patients, contributing to the lack of a neutralizing effect of anti-HDV antibodies and to the persistence of HDV infection. Certain regions, such as the NLS region, could be exposed through the dynamic interaction between HDAg and HDV-RNA and possibly reflect a regulatory mechanism within the HDV replication cycle, which needs to be confirmed by structural and functional analysis in the future. Overall, the limitations of HDAg-specific antibodies, which we clearly define in our study, need to be considered when generating therapeutic and diagnostic antibodies.

## Figures and Tables

**Figure 1 viruses-17-01220-f001:**
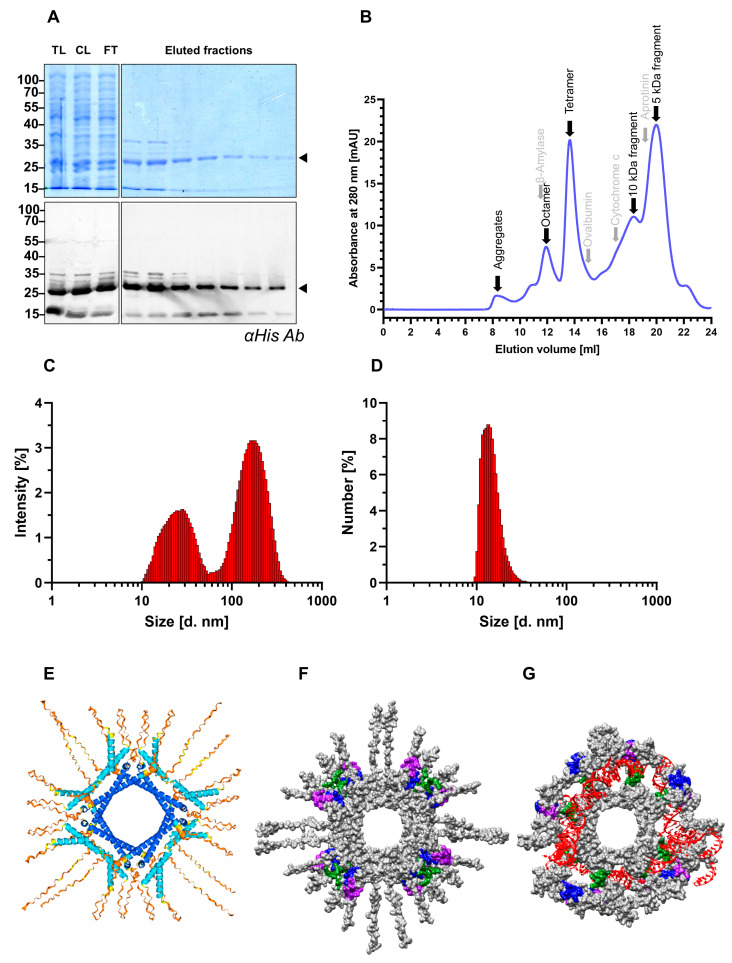
Bacterially expressed SHDAg-His_6_ is oligomeric under native purification conditions. SHDAg-His_6_ was expressed in *E. coli* and purified using Ni-NTA under native conditions. (**A**) Here, 10 µg of eluted fractions were loaded per lane of a 12% SDS-PAGE gel. The Western blot was probed with a monoclonal anti-His_6_ antibody, and the upper gel was stained with Coomassie-blue. Arrows indicate monomeric SHDAg-His_6_. TL: total lysate, CL: cleared lysate, and FT: flowthrough. (**B**) Here, 900 µg of purified SHDAg-His_6_ was loaded on a Superdex 200 Increase 10/300 GL column for analytical size exclusion chromatography. Black arrows represent retention peaks of the purified sample, in particular the presumable conformations and fragment variants of purified SHDAg-His_6_, while grey arrows indicate respective retention volumes of protein standards. Dynamic light scattering was analyzed by measuring 190 µg of purified protein sample in triplicates with 10 scans using a Zetasizer instrument. Displayed are the distribution of sizes ranging from 1 nm to 1000 nm within the purified samples as (**C**) percentage of intensity or (**D**) percentage of numbers. Structure of 8 units of SHDAg-His_6_ was predicted using Alphafold 3. (**E**) Model confidence determined by predicted local distance test (pLDDT) is indicated by color: pDLLT>90 (dark blue); 90 > pDLLT > 70 (light blue); 70 > pDLLT > 50 (yellow); pLLDT < 50 (orange). (**F**,**G**) To visualize overall HDAg conformation, the molecular surface of the prediction is colored in grey. Within the molecular surface models, the first arginine-rich motif (Lys97–Ala107) is indicated in blue, and the second arginine-rich motif (Glu136–Gly146) in purple. The nuclear localization signal is colored in green. (**G**) To visualize HDAg–protein and HDV–RNA interactions, a 394 nt long HDV genome RNA snippet (accession number: AJ000558.1; segment 197 (5′) to 1481 (3′)) depicted in red was included in the Alphafold 3 prediction.

**Figure 2 viruses-17-01220-f002:**
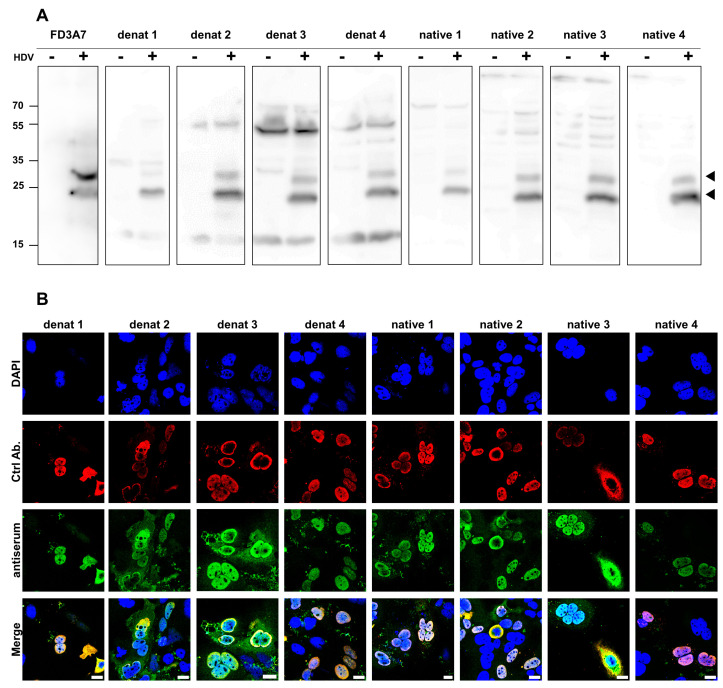
HDAg-specific antisera differ in their binding specificity under native and denaturing conditions. (**A**) HDV genome cDNA transfected Huh7 cell lysates (+) and mock transfected negative control lysates (-) were supplemented with detergent and boiled at 95 °C for protein denaturation and subsequently Western blotted. Western blots were probed with indicated HDAg-specific antisera in a 1:5000 dilution and detected using a horseradish peroxidase conjugated secondary antibody. Commercially available monoclonal anti-HDAg antibody (FD3A7) served as a reference antibody. Arrows indicate LHDAg- (upper) and SHDAg-specific (lower) protein bands. (**B**) Recombinant SHDAg-StII transfected Huh7 cells were fixed with paraformaldehyde to preserve secondary protein structure and probed with indicated HDAg-specific antisera (green). Monoclonal anti-Streptag II antibody served as a reference antibody (red). Antisera signals were detected with an Alexa488 conjugated secondary antibody, and anti-Strep-tagII signals with an Alexa594 conjugated antibody. Nuclei of all cells were stained with DAPI (blue). Untransfected cells within each sample served as a negative control. Cells were visualized with a Leica SP8 confocal laser scanning microscope. Scale bar: 20 µm.

**Figure 3 viruses-17-01220-f003:**
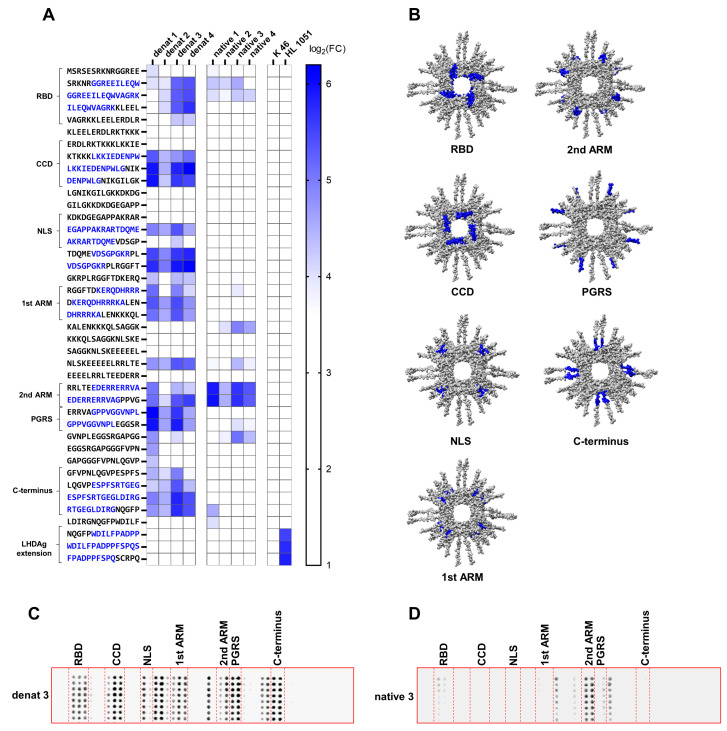
Several HDAg-specific B-cell epitopes are inaccessible in the oligomeric state of HDAg. (**A**) Peptide arrays consisting of 15 amino acid long peptides covering the full-length LDHAg sequence derived from HDV genotype 1 with 10 amino acid overlap were incubated with respective HDAg-specific antisera. Epitope recognition and binding by polyclonal antibodies within the HDAg-specific antisera were detected using a fluorophore conjugated secondary antibody. To obtain displayed log_2_ fold change (log_2_ FC), binary logarithms of respective fluorescence signals normalized to the DMSO negative control were calculated. Linear B-cell epitopes were defined by a log_2_ (FC) above 4. Functional domains are indicated in brackets alongside respective peptide sequences. Presumable epitope sequences are highlighted in blue. To validate the peptide array, HBcAg-specific antibody K46 served as a negative control, and commercially available LHDAg-specific antibody HL1051 served as a positive control. (**B**) Structural prediction of 8 units of SHDAg-His_6_ using Alphafold 3. Respective functional domains are indicated in blue. (**C**,**D**) Representative HDV genotype-1-based peptide arrays of denat 3 (**C**) and native 3 (**D**) are indicated with respective functional domains separated by dashed red lines. Each peptide array consisted of octuplicates of each peptide. Fluorescence signals of the octuplicates were averaged prior to normalization. RBD: RNA binding domain (Ser2–Leu27); CCD: coiled-coil domain (Glu31–Gly52); NLS: nuclear localization signal (Glu66–Arg75); 1st ARM: first arginine-rich motif (Lys97–Ala107); 2nd ARM: second arginine-rich motif (Glu136–Gly146); PGRS: proline-glycine-rich sequence (Gly146–Leu155); C-terminus (Glu176–Gly190).

**Figure 4 viruses-17-01220-f004:**
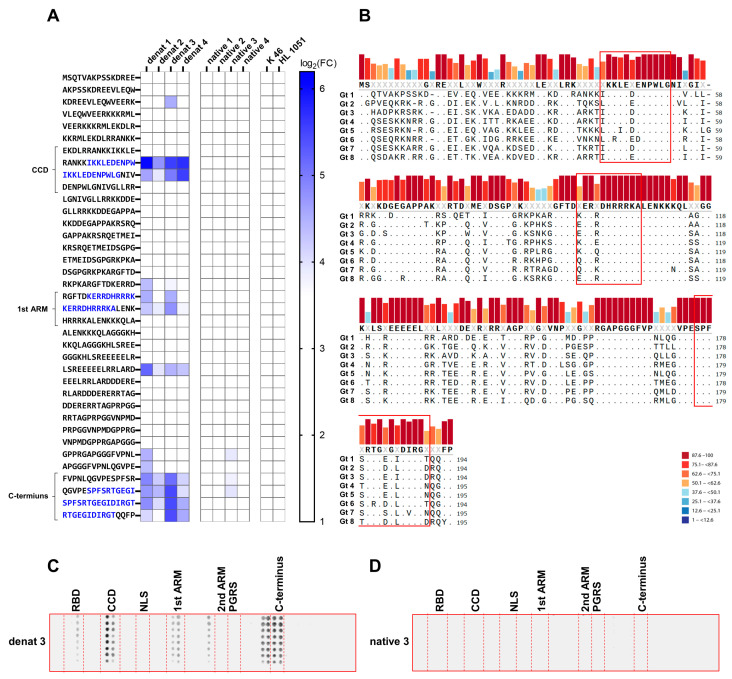
HDV-specific antisera have pan-genotypic binding capacity toward conserved epitope sequences. (**A**) Peptide arrays consisting of 15 amino acid long peptides covering the full-length SDHAg sequence derived from HDV genotype 3 with 10 amino acid overlap were incubated with respective HDAg-specific antisera. Displayed log_2_(FC) were determined as described in Figure 3. (**B**) Eight SHDAg sequences representative for the eight HDV genotypes were aligned by multiple sequence alignment. The resulting conservation scores calculated by the Valdar method are indicated as colored bars. Ranges of the scores are indicated by respective colors. Conserved residues are indicated by dots. The consensus sequence is displayed below the colored bars. The epitope sequences within the CCD, the 1st ARM, and the C-terminus are highlighted by a red box. (**C**,**D**) Representative HDV genotype-3-based peptide arrays of denat 3 (**C**) and native 3 (**D**) are indicated with respective functional domains separated by dashed red lines. RBD: RNA binding domain (Ser2–Leu27); CCD: coiled-coil domain (Glu31–Gly52); NLS: nuclear localization signal (Glu66–Arg75); 1st ARM: first arginine-rich motif (Lys97–Ala107); 2nd ARM: second arginine-rich motif (Glu136–Gly146); PGRS: proline-glycine-rich sequence (Gly146–Leu155); C-terminus (Glu176–Gly190).

**Figure 5 viruses-17-01220-f005:**
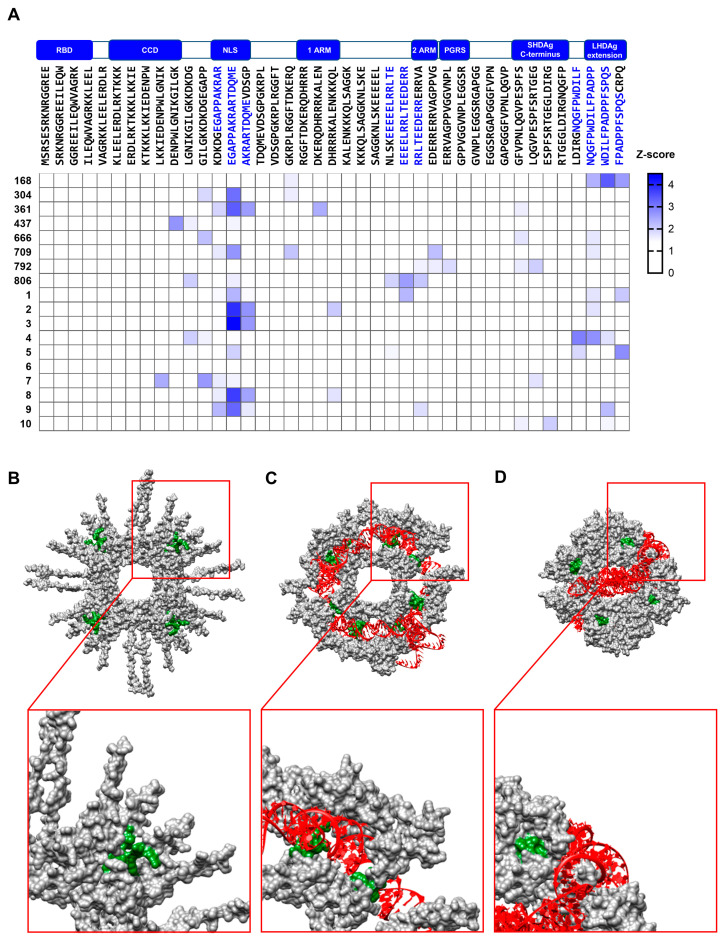
HDV-infected patient sera recognize a prominent epitope within the nuclear localization signal of HDAg. (**A**) Peptide arrays consisting of 15 amino acid long peptides covering the full-length LDHAg sequence derived from HDV genotype 1 with 10 amino acid overlap were incubated with different HDV-infected patient sera. Epitope recognition of respective sera was determined by incubation with fluorophore-coupled anti-human IgG-specific secondary antibody. Indicated Z-scores were calculated by subtracting the fluorescence intensities of the individual peptides by the average fluorescence intensity of the entire array and normalizing the difference by the standard deviation of the entire array. Locations of the functional domains are indicated above the heatmap by blue boxes. HDAg-specific linear epitopes are highlighted in blue within the peptide sequences. (**B**) To visualize conformational changes of the nuclear localization signal (NLS) region depending on the interaction with the HDV-RNA genome, eight units of SHDAg-His_6_ were predicted by Alphafold 3 either without any RNA, (**C**) with a 394 nt long HDV genome RNA snippet (accession number: AJ000558.1; segment 197 (5′) to 1481 (3′)), (**D**) or with a random 394 nt RNA sequence. The RNA sequences are colored in red and the NLS region (Glu66–Arg75) is colored in green.

**Figure 6 viruses-17-01220-f006:**
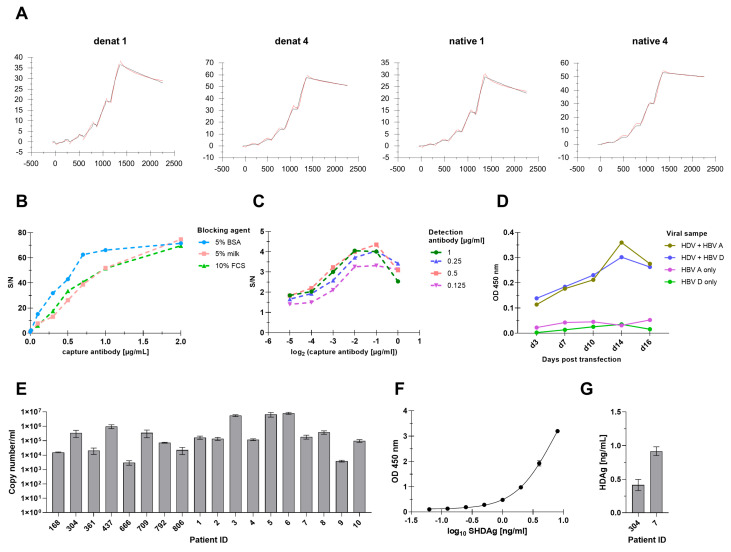
Establishment of a quantitative ELISA using highly affine HDAg-specific antibodies. (**A**) SPR measurements of HDAg-specific antisera. HDAg-specific antibodies were enriched by Protein G affinity chromatography. Then, 0.25 µg/mL purified SHDAg-StII were captured on a monoclonal anti-Strep-tag II coupled CM5 chip. A dilution series ranging from 0.0512 µM up to 2 µM was probed on the CM5 chip. (**B**) Comparison of different blocking agents for HDV-specific ELISA. ELISA plates were coated with a dilution series of purified rabbit-derived native 4 antiserum in bicarbonate buffer. Precoated plates were blocked with indicated blocking agents for 1 h at RT. Signals were detected using an anti-rabbit IgG-HRP conjugate. Displayed S/N were determined by normalizing respective signals to the negative control without a capture antibody. (**C**) Checkerboard titration of capture and detection antibodies. ELISA plate was incubated with a dilution series of purified native 4 capture antibody. Upon blocking with 5% BSA, plates were incubated with 100 ng/mL purified SHDAg-His_6_ antigen. Bound antigens were detected by incubation with a dilution series of biotinylated denat 4 antibody and subsequent incubation with a streptavidin-HRP conjugate. (**D**–**G**) Quantitative detection of HDV in vitro and in vivo by in-house ELISA. ELISA plates were coated with 0.5 µg/mL purified native 4 capture antibody in bicarbonate buffer. (**D**) Plates were subsequently either blocked with 5% BSA and incubated with supernatant samples of HDV and HBV genotypes A and D genome cDNA co-transfected or mono-transfected cells harvested at different time points (**G**) or incubated with lipid-depleted HDV-infected patient sera, which were pre-mixed with 1% BSA in TBS-T. (**F**) For quantification, plates were incubated with a dilution series of purified SHDAg-His6 diluted in 5% BSA in TBS-T. After overnight incubation of plates with respective samples at 4 °C, plates were incubated with 0.5 µg/mL biotinylated denat 4 detection antibody. Signal detection occurred by incubation with the streptavidin-HRP conjugate and TMB substrate. Absolute OD values were background subtracted. (**F**) A 4PL sigmoidal curve fit was applied to generate a standard curve based on SHDAg-His_6_ dilution series. (**G**) OD values of patient sera were interpolated to respective absolute HDAg concentrations. HDAg was detected in 2 out of 18 tested patient samples. (**E**) For validation, HDV viral RNA in patient sera was extracted and quantified by reverse-transcription qPCR.

**Table 1 viruses-17-01220-t001:** Primer sequence for generation of recombinant SHDAg constructs.

	Sequence
Primer 1	AAACATATGAGCCGGTCCGAGTCGAGGAAGAAC
Primer 2	AAAGCGGCCGCTGGAAATCCCTGGTTTCCCCTGATGTCCA
Primer 3	AAAGCGGCCGCTCATCACTTCTCGAACTGCGGGTGGCTCCACGCGCTGCCACCAGAGCCACCACCACCGGAGCCGCCACCTTTTTCAAATTGG

**Table 2 viruses-17-01220-t002:** Identified linear B-cell epitopes in the HDV HDAg protein and their accessibility.

Epitope Sequence	Region/Domain	Accessibility
GGREEILEQWAGRK	RBD	accessible in purified HDAg oligomer
LKKIEDENPWLG	CCD	inaccessible in purified HDAg oligomer
IKKLEDENPWLG *	CCD	inaccessible in purified HDAg oligomer
EGAPPAKRARTDQME	NLS	accessible in HDV in vivo inaccessible in purified HDAg oligomer
VDSGPGKR	downstream NLS	inaccessible in purified HDAg oligomer
KERQDHRRRKAL	1st ARM	inaccessible in purified HDAg oligomer
KERRDHRRK *	1st ARM	inaccessible in purified HDAg oligomer
EDERRERRVA	2nd ARM	accessible in purified HDAg oligomer
EEEEELRRLTEEDERR	upstream 2nd ARM	accessible in HDV in vivo
GPPVGGVNPL	PGRS	inaccessible in purified HDAg oligomer
ESPFSRTGEGLDIRG	C-terminus of SHDAg	inaccessible in purified HDAg oligomer
SPFSRTGEGIDIRGT *	C-terminus of SHDAg	inaccessible in purified HDAg oligomer
LDIRGNQGFP	C-terminus of SHDAg	accessible in purified HDAg oligomer
WDILFPADPPFSPQS	LHDAg extension	accessible in HDV in vivo

RBD: RNA binding domain; CCD: coiled-coil domain; NLS: nuclear localization signal; 1st ARM: first arginine-rich motif; 2nd ARM: second arginine-rich motif; PGRS: proline-glycine-rich sequence; * HDV genotype 3.

**Table 3 viruses-17-01220-t003:** Kinetic parameters of HDAg-specific antisera.

	ka (1/Ms)	kd (1/s)	KD (nM)
denat 1	1958 ± 198	2.26 × 10^−4^ ± 7.63 × 10^−5^	121 ± 51
denat 4	2182 ± 266	1.05 × 10^−4^ ± 3.08 × 10^−5^	51 ± 20
native 1	1540 ± 248	2.52 × 10^−4^ ± 5.19 × 10^−5^	173 ± 62
native 4	2492 ± 278	6.47 × 10^−5^ ± 3.00 × 10^−7^	26 ± 3

## Data Availability

All raw data used to generate the data and figures of this manuscript are available at the following link to Mendeley Data: https://data.mendeley.com/preview/h7s2kvrm66?a=abfaa40a-b826-446e-bc1d-a2c8fb988aff (accessed on 1 September 2025). The dataset will be published and the link updated upon acceptance of the manuscript.

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
