# Peer review of "Generation and Characterization of HDV-Specific Antisera with Respect to Their Application as Specific and Sensitive Research and Diagnostic Tools"

_viruses, 2025, doi:10.3390/v17091220_

Round 1
Reviewer 1 Report
Comments and Suggestions for Authors
The manuscript by Thiyagarajah et al., “Generation and characterization of HDV-specific antisera with respect to their application as specific and sensitive research and diagnostic tools” presents a study on the generation, structural characterization, and diagnostic application of polyclonal antibodies targeting the hepatitis D virus (HDV) small delta antigen (SHDAg). The authors also attempt to bridge in vitro antibody generation with clinically relevant epitope mapping in human patient sera. The study addresses an important need for cost-effective HDV diagnostics, especially in resource-limited settings, where RT-qPCR may not be practical. The work is promising and addresses a significant diagnostic challenge for HDV. However, there are some issues that must be addressed before publication.
In abstract, the claim “These antibodies were used to establish a highly sensitive ELISA that can quantitatively detect HDV virions and thus can be used as an alternative diagnostic screening method” is not sustained by the results. Although the proposed ELISA demonstrate robust in vitro detection, it detected SHDAg in only 2 of 18 HDV RNA–positive patient sera.
Abbreviations (e.g., SPR) should be defined in full the first time they appear.
The order of the Methods section should be revised. Information on patient sera and ethics should be presented before Section 2.10, where sera are first used. In addition, Section 2.11 describes the purification of native4 and denat4, but their selection is only justified in Section 2.12.
Section 2.1 mentions pSVLD3 , while Section 2.7 refers to pcDNA3.1 for SHDAg-StII. It is unclear if both plasmids were used for different experiments.
Protein purification for SPR analysis is described twice, in sections 2.1 and 2.11
Section 2.8, the type of control antibodies is not specified (FD3A7 is the clone). Also, in some cases in methods, the type of secondary antibodies is not specified.
In the Western blot analysis, the molecular weight marker used is not specified.
In section 2.9, the origin of the Huh7 cells is not specified, and cell culture conditions are not described. In addition, AlexaFluor 488 is a fluorochrome and the conjugated antibody should be clearly identified.
In section 2.16, the type of serial dilutions used for generating the standard curve in the ELISA assay is not specified. Also, criteria for defining positive vs. negative signals are not stated. Moreover, no sera from patients with HBV-infection only were included as controls to test assay specificity. This omission is critical, since HDV infection always occurs in the context of HBV infection, and HBV-related antigens could confound the ELISA results.
The SHDAg-His6 preparation was only ~80% pure and contained degradation products. While the authors attribute these to truncated forms of SHDAg, this impurity could affect epitope presentation and antibody specificity.
The figure legends include details that are not described in the Methods section (e.g., Section 2.10 vs. Figure 3: negative and positive controls).
It is unclear how the appendix connects to the main text. While it provides additional methodological details that could be integrated into the main manuscript, it also introduces divergent information (e.g., the use of the pET21a(+) expression vector for plasmid construction in A.1, which is inconsistent with the plasmid description in the main article). Moreover, the appendix is not cited anywhere in the main text.
Author Response
Point-by-point reply to Reviewer 1
- In abstract, the claim “These antibodies were used to establish a highly sensitive ELISA that can quantitatively detect HDV virions and thus can be used as an alternative diagnostic screening method” is not sustained by the results. Although the proposed ELISA demonstrate robust in vitro detection, it detected SHDAg in only 2 of 18 HDV RNA–positive patient sera.
- We agree, that this statement is misleading, since it is not fully sustained by the results. We now rewrote the statement, emphasizing more that it is suitable for quantitative HDV virion detection in vitro and upon further optimization can also be applied in HDV diagnosis (see lines 34-36).
- Abbreviations (e.g., SPR) should be defined in full the first time they appear.
- As suggested by the reviewer, we revised the entire manuscript and defined all abbreviations in full the first time they appear in the main text.
- The order of the Methods section should be revised. Information on patient sera and ethics should be presented before Section 2.10, where sera are first used. In addition, Section 2.11 describes the purification of native4 and denat4, but their selection is only justified in Section 2.12.
- As requested by the reviewer we changed the order of the methods reviewer for better clarity and readability.
- Section 2.1 mentions pSVLD3, while Section 2.7 refers to pcDNA3.1 for SHDAg-StII. It is unclear if both plasmids were used for different experiments.
- We apologize for the confusion and clarified this point in the revised version of the manuscript. The SHDAg-His6 plasmid was constructed using pSVLD3 as template while the SHDAg-StII encoding prokaryotic expression vector was constructed using the previously described eukaryotic expression vector pCDNA3.1(-)-SHDAg-StII. This particular expression vector was also used for transfection of the mammalian Huh7 cells for the immunofluorescence analysis. To avoid confusion, we included the section “2.1 plasmid construction”, in which we describe the construction of both bacterial expression vectors and respective templates (lines 101-111). We also included the respective plasmid constructs used in the specific experiments (see lines 114-116; 174-178 and 180-182).
- Protein purification for SPR analysis is described twice, in sections 2.1 and 2.11
- We clarified this point in the revised version. Initially, section 2.1 described the purification of the antigen SHDAg-StII, while section 2.11 refers to the HDAg specific antibodies within the antisera, which were purified by protein G affinity chromatography. To avoid any confusion, we rewrote the specific section clarifying the purification of the antigen on the one hand and of the antibodies on the other hand (lines 259 – 276).
- Section 2.8, the type of control antibodies is not specified (FD3A7 is the clone). Also, in some cases in methods, the type of secondary antibodies is not specified.
- As suggested by the reviewer we included more details for all secondary antibodies used in this study (see lines 195 – 201 and 209 – 213).
- In the Western blot analysis, the molecular weight marker used is not specified.
- As requested by the reviewer we added the molecular weight marker used for the Western blot analysis in the methods section in line 191.
- In section 2.9, the origin of the Huh7 cells is not specified, and cell culture conditions are not described. In addition, AlexaFluor 488 is a fluorochrome and the conjugated antibody should be clearly identified.
- As kindly pointed out by the reviewer, we included the necessary details for the cells and antibodies (see lines 170-173; 209 – 213).
- In section 2.16, the type of serial dilutions used for generating the standard curve in the ELISA assay is not specified. Also, criteria for defining positive vs. negative signals are not stated. Moreover, no sera from patients with HBV-infection only were included as controls to test assay specificity. This omission is critical, since HDV infection always occurs in the context of HBV infection, and HBV-related antigens could confound the ELISA results.
- We see this point, that certain necessary information regarding the ELISA are missing and modified the manuscript to address this relevant point. We therefore expanded the section 2.16 in the revised version of out manuscript describing the ELISA with the necessary information (see lines 311 – 328). We tested supernatant samples of HBV genotype A and D monotransfected cells. These supernatant samples contain high amounts of infectious HBV particles, HBV subviral particles as well as HBV capsids. Since the HBV supernatant samples did not show any signals, assay interreference by HBV viral components can be ruled out. Therefore, the HBV supernatant samples were used as negative control samples defining the lower threshold within each ELISA run. This information is also included in lines 325 – 328 of the revised version. Western blot analyses of lysates derived from HBV expressing cells or immunofluorescence microscopy of HBV-positive cells using these sera confirmed that these sera have no cross reactivity with HBV-specific proteins.
- The SHDAg-His6 preparation was only ~80% pure and contained degradation products. While the authors attribute these to truncated forms of SHDAg, this impurity could affect epitope presentation and antibody specificity.
- We see the point, that the degradation products in our antigen preparation could have affected epitope presentation and antibody specificity. In preparatory experiments various conditions were tested to avoid SHDAg degradation during protein production and purification. However, it was not possible to fully avoid protein degradation. Since HDAg degradation is also observed in the liver tissue of HDV infected woodchucks and humans, as described Jia-Gang Wang et al., 1992
https://doi.org/10.1099/0022-1317-73-1-183 and Michael Roggendorf et al., 1987 https://doi.org/10.1099/0022-1317-68-11-2953
we assume that the observed degradation is an intrinsic characteristic, probably an autoproteolytic property of the SHDAg protein, and thus inevitable. Nevertheless, to emphasize more on this very important fact, we included a statement, in which we discuss these considerations (see lines 730-737)
- The figure legends include details that are not described in the Methods section (e.g., Section 2.10 vs. Figure 3: negative and positive controls).
- As suggested by the reviewer we included the information provided in the figure legends also in the materials and methods section in lines 238-241.
- It is unclear how the appendix connects to the main text. While it provides additional methodological details that could be integrated into the main manuscript, it also introduces divergent information (e.g., the use of the pET21a(+) expression vector for plasmid construction in A.1, which is inconsistent with the plasmid description in the main article). Moreover, the appendix is not cited anywhere in the main text.
- As suggested by the reviewer, we integrated the additional information of the appendix into the materials and methods section of the main text and removed the appendix. We also revised the entire materials and methods section and corrected all inconsistencies (lines 98 – 333)

Reviewer 2 Report
Comments and Suggestions for Authors
This is a scientifically relevant and well-executed study addressing an important unmet need in HDV diagnostics. The methodology is solid, the experimental work is comprehensive, and the results are clearly presented. The manuscript offers valuable insights into antibody generation and characterization against SHDAg. However, some statements, particularly regarding the diagnostic applicability of the new ELISA developed, are somewhat overstated based on the current data. Clarifications and additional patient-related information would strengthen the work.
Please find below my specific comments:
1) The Introduction, while informative, is overly detailed in the virological background of HDV. I recommend shortening this section to focus more directly on the rationale for generating HDAg-specific antibodies and the unmet diagnostic need.
2) Both the Abstract and the Introduction currently overstate the impact of the ELISA assay (“highly sensitive”, “alternative diagnostic tool”, etc.). Given that only 2 out of 18 HDV RNA-positive patients samples tested positive by ELISA, the diagnostic potential of the assay needs to be presented more cautiously.
3) The manuscript currently does not provide any clinical, biochemical or virological information on the HDV-infected patients whose sera were tested. I strongly recommend adding a table summarizing the main demographic, virologic, biochemical and clinical characteristics of the patients included in the study (eve as supplementary table).
4) Again, the phrasing “we established an ELISA that enables the quantitative detection of HDAg within HDV virions and can be implemented in clinics as an alternative diagnostic method” (lines 667-671 of the discussion section) appears too strong. Please consider softening this statement to acknowledge the assay’s current limitations, especially in patient samples.
5) While the ELISA has shown limited performance in patient sera, the concept of quantitative HDAg detection remains valuable. I encourage the authors to expand the discussion on potential clinical applications, particularly in light of 1) the limited availability and standardization of HDV RNA assays worldwide, 2) modest reproducibility of HDV RNA quantification, as shown in the external quality studies (consider citing PMID: 40784197), and 3) the possible role of HDAg as a complementary biomarker for monitoring therapy or viral clearance.
Author Response
Point-by-point reply to Reviewer 2
- The Introduction, while informative, is overly detailed in the virological background of HDV. I recommend shortening this section to focus more directly on the rationale for generating HDAg-specific antibodies and the unmet diagnostic need.
- We agree with the reviewer. Accordingly, we shortened the introduction and highlighted the rationale of this study more in depth (see lines 40 - 97).
- Both the Abstract and the Introduction currently overstate the impact of the ELISA assay (“highly sensitive”, “alternative diagnostic tool”, etc.). Given that only 2 out of 18 HDV RNA-positive patients samples tested positive by ELISA, the diagnostic potential of the assay needs to be presented more cautiously.
- We see this point, that at this stage, the ELISA for testing patient samples needs further optimization and conclusions regarding its applicability as diagnostic tools need to be stated more cautiously. Therefore, we rephrased the part regarding the ELISA in the abstract and the introduction. We have therefore described the potential of ELISA as a possible future diagnostic tool in a much more cautious manner (see lines 34-36 and 92 – 97).
- The manuscript currently does not provide any clinical, biochemical or virological information on the HDV-infected patients whose sera were tested. I strongly recommend adding a table summarizing the main demographic, virologic, biochemical and clinical characteristics of the patients included in the study (eve as supplementary table).
- As suggested by the reviewer we included table A1 “Demographic and clinical characteristics of HDV infected patients” as a supplementary table in which all available patient characteristics are listed. The table is also mentioned in lines 802-803.
- Again, the phrasing “we established an ELISA that enables the quantitative detection of HDAg within HDV virions and can be implemented in clinics as an alternative diagnostic method” (lines 667-671 of the discussion section) appears too strong. Please consider softening this statement to acknowledge the assay’s current limitations, especially in patient samples.
- As pointed out by the reviewer, we carefully modified the particular statement regarding the ELISA of patient samples and its diagnostic application by emphasizing that it has potential as an alternative but needs further optimization (see lines 725-728).
- While the ELISA has shown limited performance in patient sera, the concept of quantitative HDAg detection remains valuable. I encourage the authors to expand the discussion on potential clinical applications, particularly in light of 1) the limited availability and standardization of HDV RNA assays worldwide, 2) modest reproducibility of HDV RNA quantification, as shown in the external quality studies (consider citing PMID: 40784197), and 3) the possible role of HDAg as a complementary biomarker for monitoring therapy or viral clearance.
- We suggested by the reviewer we modified the discussion and put more emphasis on the usage of HDAg as a complementary biomarker and on the high potential of ELISA as an alternative diagnostic tool, also in light of the limitations of current HDV RNA quantification assays. We included a statement in lines 817-825 emphasizing on these considerations.

Round 2
Reviewer 1 Report
Comments and Suggestions for Authors
The authors have addressed the main comments and improved the manuscript. However, a few minor corrections are still required: at present, there are two different tables labeled as “Table 1” - one in the Methods section (page 4) and another in the Results section (page 16). To avoid confusion, please adjust the numbering. In addition, the primer table would be more appropriately placed immediately after section 2.1, where the primers are first mentioned.
Author Response
Reviewer...."there are two different tables labeled as "Table 1"...:
Answer: We apologize for this mistake and corrected this point. In accordance to this we adjusted the subsequent numbering of the tables and their mention in the text.
Reviewer: ....the primer table would be more appropriately placed immediately after section 2.1
Answer: as requested by the reviewer we placed table 1 immediately after section 2.1
